# Machine Learning Model for Leak Detection Using Water Pipeline Vibration Sensor

**DOI:** 10.3390/s23218935

**Published:** 2023-11-02

**Authors:** Suan Lee, Byeonghak Kim

**Affiliations:** School of Computer Science, Semyung University, Jecheon 27136, Republic of Korea; hahaja01@semyung.ac.kr

**Keywords:** water leak detection, deep learning, machine learning, time-frequency analysis

## Abstract

Water leakage from aging water and wastewater pipes is a persistent problem, necessitating the improvement of existing leak detection and response methods. In this study, we conducted an analysis of essential features based on data collected from leak detection sensors installed at water meter boxes and water outlets of pipelines. The water pipeline data collected through the vibration sensor were preprocessed by converting it into a tabular form by frequency band and applied to various machine learning models. The characteristics of each model were analyzed, and XGBoost was selected as the most suitable leak detection model with a high accuracy of 99.79%. These systems can effectively reduce leak detection and response time, minimize water waste, and minimize economic losses. Additionally, this technology can be applied to various fields that utilize water pipes, making it widely applicable.

## 1. Introduction

The detection of leaks is a crucial issue in today’s world, as it has the potential to affect the environment, human safety, and result in economic losses. Leaks of water, gas, electricity, and oil can occur in various forms, and hence, the detection of leaks is crucial in industrial, commercial, and residential areas. The objective of leak detection is to detect and control leaks in a timely manner to create a safe environment and minimize losses. Leak detection technologies can be broadly classified into physical and electronic methods. Physical methods are traditional and require experienced professionals, making them labor-intensive, time-consuming, and costly. Electronic methods, on the other hand, use modern technologies such as sensor technology, data analytics, and artificial intelligence (AI) to detect and analyze leaks. Therefore, electronic leak detection using AI is receiving a lot of attention from researchers.

AI-based leak detection systems can quickly determine the source and location of a leak by analyzing data collected from various sensors and suggesting the best course of action to resolve it. IoT technology can be utilized to monitor leaks in real-time and respond automatically in conjunction with a centralized control system. However, there are still significant challenges to be addressed in the development of leak detection technology, such as improving the accuracy and sensitivity of detection sensors, ensuring performance in various environments, and developing efficient data processing and analysis methods. Additionally, aging water pipelines consisting of a mixture of metal and non-metal pipes, coupled with various noises, including environmental, electrical, and natural sounds, make it difficult to detect leaks. There is an urgent need to develop technology that can accurately detect leaks in such a complex environment. In this paper, we propose a leak detection model that can overcome these challenges and show high performance in various environments. The proposed model is designed to address the aging problem and various noises in water pipes with a mixture of metal and non-metal pipes. We compare the proposed model with representative machine learning and deep learning models such as K-Nearest Neighbor, decision tree, random forest, extra trees, LightGBM, XGBoost, and CatBoost. Our results show that the XGBoost model achieved the highest performance with an accuracy of 99.79%.

The key contributions of the paper are

Introduction of a Novel Leak Detection Model: The paper proposes a new leak detection model specifically designed to address challenges associated with aging water pipelines made of a combination of metal and non-metal materials. This model also effectively handles various types of noise, including environmental, electrical, and natural sounds, which can interfere with leak detection.Comprehensive Model Comparison: The proposed model was rigorously compared with several representative machine learning models, including K-Nearest Neighbor, decision tree, random forest, extra trees, LightGBM, XGBoost, and CatBoost.Applicability in Diverse Environments: The results underscore the model’s capability to accurately detect and pinpoint leaks in water pipelines that comprise a mix of different materials and are subjected to various noises. This is crucial for the efficient management and upkeep of water supply systems.

This paper is organized to provide detailed information about leak detection data and machine learning models for it. The paper begins with an “Introduction” and reviews the existing literature in the field in the “Related Works” section. Section 3, “Water Leakage Detection Framework”, systematically explores the data collected from water pipeline vibration sensors, preprocessing, and different machine learning models. The “Experimental Results” section details the detailed evaluation of the models across multiple experiments. The paper concludes with a “Conclusions” that summarizes the main results and implications of this study.

## 2. Related Works

Recently, advancements in sensor technology have led to the proliferation of diverse sensors, enabling efficient data acquisition via sensor networks [1,2]. Specifically, this section aims to review the extant literature pertaining to leak detection methodologies. This paper reviews water pipeline leakage detection techniques, which can be classified into three categories: software-based, hardware-based, and conventional methods; it then presents a comparative study of vibration sensors for water pipeline leakage detection and validates a water pipeline testbed using vibration sensors [3]. Liu et al. propose a water pipeline leakage detection method based on machine learning and wireless sensor networks (WSNs) that employs a leakage triggered networking method to reduce energy consumption and a leakage identification method using intrinsic mode function (IMF), approximate entropy (ApEn), principal component analysis (PCA), and a support vector machine (SVM) to enhance the precision and intelligence of leakage detection [4]. Fereidooni et al. propose a fast hybrid method using AI algorithms and hydraulic relations for detecting and locating leaks and identifying the volume of losses material in large scale water distribution networks (WDN) [5]. Luong et al. propose a data renovation method to improve the generalization ability of training data for an intelligent leak detection system based on statistical parameters extracted from acoustic emission signals [6]. Nkemeni et al. present a fully distributed solution for leak detection in a water distribution network using a distributed Kalman filter (DKF) that improves the accuracy of leak detection and power consumption in WSN applications [7]. Shukla et al. present a deep learning algorithm that uses scalogram images of vibration signals collected from accelerometers attached to the pipeline surface to detect leakages in water pipelines with up to 95% accuracy [8]. Mysorewala et al. present a feasibility study of leak detection in wall-mounted water pipelines through vibration measurements using low-power accelerometers; they offer a cost-effective and energy-efficient scheme to detect and classify leaks by optimally placing sensor nodes at carefully selected locations [9].

Wang et al. present an experimental study on water pipeline leak detection using in-pipe acoustic signal analysis and artificial neural network (ANN) prediction to investigate the effects of leak size, pipeline pressure, and flow velocity on the characteristic of acoustic signal and to improve the accuracy of leak recognition [10]. Guo et al. propose a time–frequency convolutional neural network (TFCNN) model for detecting leaks in water distribution systems based on acoustic signals, which improves the accuracy and stability of leakage detection even under low signal-to-noise ratio conditions [11]. Ravichandran et al. present an acoustic leak detection system for distribution water mains using machine learning methods, specifically a multi-strategy ensemble learning approach, which has demonstrated significant improvement in performance, resulting in a reduction of false positive reports by an order of magnitude [12]. Zhou et al. propose a novel ensemble transfer learning one-dimension convolutional neural network (TL1DCNN) approach for pipeline leak detection and localization, which integrates the results of a set of base learners to achieve superior performance compared to traditional methods and other deep learning methods [13]. Liu et al. describe a novel approach to leak detection in water pipes using a Maximum Entropy version of the Least Square Twin K-Class Support Vector Machine (MLT-KSVC) algorithm. This approach assigns different weights to leak samples based on the MaxEnt model, reducing the impact of outliers on the classification process and improving accuracy compared to other methods [14].

Pipelines are one of the least expensive means of transporting fluids over long distances and distributing fluids in large areas and cities. As such, monitoring these pipelines to predict and detect leakage accurately and promptly and to determine the location of the leak is of importance. Sekhavati et al. provides a review and comparative study of computational methods for pipeline leakage detection and localization, discussing the strengths, weaknesses, and limitations of five types of methods: mass/volume balance, negative pressure wave, pressure point analysis, statistical methods, and real-time transient modeling [15]. Tariq et al. present a study on the application of cost-effective MEMS-based accelerometers for leak detection in real water distribution networks, where experiments were conducted over ten months, and machine learning models were developed to improve leak detection accuracy [16]. Tijani et al. propose a reliable technique for pipeline leak detection using acoustic emission signals and deep learning to extract leak-related discriminant features from acoustic images obtained from time series acoustic emission signals using continuous wavelet transform [17]. Ahmad et al. propose a reliable technique for pipeline leak detection using acoustic emission signals and deep learning to extract leak-related features from acoustic images obtained from time series acoustic emission signals using continuous wavelet transform [18]. Xu et al. propose a method for identifying leaks in water pipes using an explainable ensemble tree model of vibration signals based on the wave propagation model and the leakage noise mechanism [19].

Deep learning techniques and algorithms are emerging as a disruptive technology with the potential to transform global economies, environments, and societies. Fu et al. provide a critical review of the role of deep learning in urban water management, examining its current applications and potential future directions to address key challenges in the field [20]. Yu et al. present a study on the effectiveness and practicability of using machine learning models to identify leaks in real pipe networks by classifying vibration signals collected by piezoelectric accelerometers installed in water distribution systems over several cities of China [21]. Zhang et al. describe the development of a convolutional neural network (CNN)–based model to classify acoustic wave files collected by the South Australian Water Corporation’s (SA Water’s) smart water network (SWN) over the city of Adelaide for pipe leak and crack detection with an accuracy of 92.44% [22]. Vanijjirattikhan et al. present the development of an AI-based water leak detection system with cloud information management that can systematically collect and manage leakage sounds and generate a model used by a mobile application to provide operators with guidance for pinpointing leaking pipes [23]. Choudhary et al. present a novel 1-D convolution neural network (1DCNN) model for leak detection, location, and size estimation in a smart water grid (SWG) that uses IoT sensors and devices to monitor water transportation; their method showed better accuracy compared to other state-of-the-art machine learning techniques [24].

Water leakage in the supply system is a silent problem that costs billions of dollars yearly. This paper is a systematic review of forty-seven articles on water leakage detection and location research, with the aim of identifying new technology, trends, and possible future directions in the field [25]. Shen et al. present a study on a tree-based machine learning method for pipeline leakage detection in water distribution systems, where the authors develop and compare three machine-learning-based models using on-site leak detection signals; they find that the AdaBoost model had the lowest false positive rate, and the recall rates of the random forest and AdaBoost models were 100% and 99.52%, respectively [26]. Choi et al. propose a convolutional neural network (CNN) model to detect and classify water leakage in pipelines using vibration data collected by leakage detection sensors, demonstrating superior performance over a support vector machine model in terms of F1-score and Matthew’s correlation coefficient [27]. Yussif et al. present a study that proposes a low-cost approach to locating leakages in urban water distribution networks using acoustic signal behavior and machine learning, achieving high validation accuracy with the developed models [28]. Ullah et al. propose a machine-learning-based platform for detecting pipeline leaks of various pinhole sizes using acoustic emission sensor channel information and achieves an exceptional overall classification accuracy of 99% [29]. 

## 3. Water Leakage Detection Framework

In contemporary industrial settings, machine learning models have become indispensable tools, finding applications across a multitude of domains. Within the ambit of this paper, we undertake a systematic comparative analysis of several machine learning models with the specific aim of identifying the most adept model for leak detection. The procedural flow of our proposed methodology is illustrated in Figure 1.

In our study, a comprehensive set of 512 features was distilled from the data amassed through vibration sensors affixed to the water pipelines, leveraging time-frequency domain feature extraction techniques. We incorporated several features for model training, which included the site number, sensor number, and date of leak detection, all of which serve as identification information. Additionally, aggregate values such as leak rate and leak level were introduced as features. Furthermore, we utilized the maximum number of leak detections, represented across 20 columns, as an additional feature. Proceeding to the next phase, we juxtaposed the performance of eight distinct machine learning models. In the concluding phase, these models underwent a rigorous evaluation, validation, optimization, and hyperparameter tuning process to ascertain their efficacy and precision.

### 3.1. Water Pipeline Leak Vibration Dataset

Water pipeline leak detection is a critical aspect of water infrastructure management. The timely detection of leaks can help prevent water loss, reduce repair costs, and prevent damage to infrastructure. However, traditional leak detection methods can be time-consuming and costly. As a result, there is a growing interest in the development of machine learning models for water pipeline leak detection. To develop accurate and reliable machine learning models, it is important to have high-quality training data. The water pipeline leak detection dataset provides a valuable resource for developing such models. However, the dataset may contain biases that can affect the performance of the models. 

To address this issue, domain experts examined data bias through time-frequency analysis and clustering of vibration detection sensor data. Time-frequency analysis is a signal processing technique used to analyze non-stationary signals in the time-frequency domain. Clustering is a machine learning technique used to group similar data points together. The domain experts used these techniques to identify patterns and structures in the data that may not be visible to the naked eye, which helped reduce data bias and improve the accuracy of the machine learning models. In addition, leak detection experts identified and labeled leak sounds through precise reading of leak points. This process provided accurate labeling information for each case in the dataset, which is essential for developing machine learning models for water pipeline leak detection.

The dataset used in this paper is water pipe leak vibration data, which consists of 30,000 cases using leak detection sensors installed at more than 11,000 locations. The dataset comprises various sounds related to outdoor leak, indoor leak, electric noise, other noise, and normal noise. The judgment criterion for water leakage is a sensor installed at the water outlet that detects water leakage, and a water leakage detection specialist will investigate the leakage on site. If a leak occurs outdoors, it is recorded as an ‘outdoor leak’, and if a leak occurs indoors, it is recorded as an ‘indoor leak’. If there is no sign of a leak, but certain noises such as mechanical or electrical sounds are generated, it is recorded as ‘Electric noise’, and if other types of noise are generated, it is recorded as ‘Other noise’. If no other leak is detected, it is recorded as ‘Normal noise’. The acquisition of data is based on water pipeline leak vibration data, followed by confirmation of the acquired data through leak detection, data refinement, and classification by class. The refined data include labeling information for each class, as shown in Table 1. This labeling information provides a valuable resource for developing machine learning models for water pipeline leak detection.

There are various types of leaks that can occur in water supply systems, and it is important to identify and categorize them accurately. Table 2 contains a detailed description of the water pipeline leak detection data. The dataset includes various items such as site number, sensor number, leakage vibration size detected by frequency, maximum detection frequency, and maximum detection size.

### 3.2. Leak Data Analysis and Preprocessing

In this paper, the dataset utilized for analysis comprises sensor information, vibration data with a frequency of 10 Hz, and aggregate data. The vibration data with a frequency of 0 Hz were excluded as they possessed identical values. The sensor information was preprocessed based on the model characteristics, and ten leak detection checks were conducted for two hours each day during the early morning hours. The average value and aggregate value of the size that responded above a specific threshold were included as features. Figure 2 presents the vibration data visualized by five classification criteria, while Figure 3 illustrates the max 1–19 values, representing the highest frequency and maximum leak size detected in one leak detection. The visualization presented in Figure 4 pertains to the correlation analysis of the vibration data collected at a frequency of 10 Hz.

### 3.3. Leak Detection Models

#### 3.3.1. K-Nearest Neighbor

The K-Nearest Neighbor (KNN) algorithm is a non-parametric, instance-based learning methodology used for classification and regression tasks [30]. Rooted in the principle of similarity, KNN determines an input instance’s output based on the majority label or mean value of its ‘K’ most similar instances from the training dataset. The algorithm’s simplicity is a significant advantage, requiring no explicit training phase. However, KNN’s computational complexity increases linearly with the size of the training dataset, making it less ideal for large datasets. Despite its simplicity, KNN can achieve high accuracy in scenarios where decision boundaries are irregular. Given its instance-based nature, KNN inherently supports multi-class classification. In this paper, a KNN model was used to interpret and analyze water and sewer vibration sensor data.

#### 3.3.2. Decision Tree

Decision trees are a popular machine learning algorithm that can be used for both classification and regression tasks [31]. The algorithm works by recursively partitioning the input space into smaller subsets as shown in Figure 5, based on the value of the input features, until a stopping criterion is met. The decision to split a node based on a feature j and threshold t can be represented using a split criterion. For classification, the Gini impurity is commonly used:(1)$$Gini\left(D\right)=1-\sum_{i=1}^{k}p_{i}^{2}$$
where D represents the data subset at the node, $p_{i}^{2}$ is the proportion of samples of class i in D, and k is the number of classes. The aim during the split is to find a feature and threshold that minimizes the weighted average of the Gini impurity of the child nodes. Decision trees have several advantages, including their interpretability, ease of use, and ability to handle both numerical and categorical data. In this paper, we used the decision tree model for the interpretation and analysis of water and sewerage vibration sensor data in a table structure. 

#### 3.3.3. Random Forest and Extra Trees

Random forest is an ensemble learning method that combines multiple decision trees to improve their performance [32]. It works by constructing a large number of decision trees, each trained on a different random subset of the training data and a random subset of the input features. The outputs of the individual trees are then aggregated to make a final prediction as shown in Figure 5. The key idea behind random forest is that by aggregating the outputs of multiple decision trees, the overall prediction becomes more robust and less prone to overfitting. For a classification problem with N classes, the random forest output, Y, for an input vector X is
(2)$$Y\left(X\right)=arg\;\underset{j}{\max}\sum_{i=1}^{T}I\left(h_{i}\left(X\right)=j\right)$$
where T is the number of trees in the forest, and $h_{i}\left(X\right)$ is the class predicted by the $i^{th}$ tree. $I\left(\cdot \right)$ is the indicator function, which is 1 if the condition inside is true, and 0 otherwise. Random forest has several advantages over single decision trees, including improved accuracy, robustness to noise and outliers, and the ability to handle high-dimensional data. In this paper, we use a random forest model to experiment with the importance of attributes and the efficiency of parallelization.

Extremely randomized trees (extra trees) [33] represent a form of ensemble learning technique grounded on decision trees, drawing parallels to the well-established random forest algorithm. The extra trees classifier constructs a multitude of decision trees and amalgamates their outputs to enhance prediction accuracy. Distinctively, in contrast to the random forest, extra trees introduce a greater degree of randomization in the selection of node splits. This distinct feature facilitates the generation of a more heterogeneous set of trees, thereby augmenting the model’s robustness and capacitating it to encapsulate a broader spectrum of characteristics.

#### 3.3.4. Gradient Boosting

Gradient boosting is an ensemble machine learning technique that seeks to optimize a differentiable loss function through iterative refinement of predictions [34]. It builds an additive model in a stage-wise fashion, where each subsequent model corrects the errors of its predecessor. At each iteration, it fits a decision tree to the negative gradient (residual errors) of the loss function. Given a dataset $\left\{\left(x_{1},y_{1}\right),\ldots ,\left(x_{N},y_{N}\right)\right\}$, where $y_{i}$ can be 0 or 1, the prediction of the ensemble at iteration m for an input vector x in terms of log odds is
(3)$$F_{m}\left(x\right)=F_{m-1}\left(x\right)+v\cdot h_{m}\left(x\right)$$
where $F_{m-1}\left(x\right)$ is the prediction of the ensemble up to the ${\left(m-1\right)}^{th}$ model, $h_{m}\left(x\right)\;$ is the prediction of the $m^{th}$ tree, and v is the learning rate. The central idea revolves around strengthening a weak learner, typically a decision tree, into a robust model by aggregating the outcomes of several trees. Hyperparameters, such as learning rate and tree depth, play pivotal roles in controlling overfitting and the algorithm’s speed. Unlike random forest, which builds trees in parallel, gradient boosting builds trees sequentially. We used the gradient boosting method for leak detection as a basic model for boosting methods.

#### 3.3.5. LightGBM

LightGBM is one of the implementations of the Gradient Boosting Decision Tree (GBDT) algorithm developed by Microsoft. It shows high performance on large datasets and has faster training speed and lower memory usage compared to other GBDT implementations. This performance improvement is possible because LightGBM uses various optimization techniques [35].

The key idea of LightGBM is to divide the entire dataset into small datasets called “leaves” to efficiently process them. LightGBM constructs trees based on these leaves. LightGBM uses a leaf-wise method, which selects the leaf node with the largest information gain when determining the tree structure as shown in Figure 6. LightGBM also supports both feature parallelism and data parallelism. Feature parallelism is a method of creating multiple trees for one dataset by using different features for each tree. In this paper, we used the LightGBM model, which provides high efficiency and fast processing speed using table-structured water and sewer vibration sensor data.

#### 3.3.6. eXtreme Gradient Boosting (XGBoost)

XGBoost, which stands for eXtreme gradient boosting, is a popular distributed gradient boosting library introduced by Tianqi Chen in 2014 [36]. The library extends the traditional gradient boosting algorithm by incorporating overfitting regularization, enabling it to process large amounts of data accurately and quickly. XGBoost can effectively solve various problems through parallel tree boosting as shown in Figure 7, making it a widely used algorithm in machine learning competitions and real-world applications. The prediction of the ensemble at iteration m for an input vector x is
(4)$${\hat{y}}_{i}^{\left(m\right)}=\;\sum_{k=1}^{m}f_{k}\left(x_{i}\right)$$
where ${\hat{y}}_{i}^{\left(m\right)}$ is the predicted value for observation i at iteration m. $f_{k}\left(x\right)$ is the prediction of the $k^{th}$ tree. The model is highly customizable, allowing users to specify various hyperparameters to optimize model performance.

In this study, we utilized the fast learning speed and high performance of the XGBoost model to train it for water and sewage leak detection. The XGBoost algorithm was employed to analyze a large dataset of water and sewage system data, which included a variety of parameters such as flow rates, pressure readings, and other key indicators of system performance. Through the use of XGBoost, we were able to effectively identify patterns and anomalies in the data that could indicate the presence of leaks or other issues in the system.

#### 3.3.7. CatBoost

CatBoost, developed by Yandex, is a state-of-the-art gradient boosting algorithm designed explicitly for efficiently handling categorical features without the need for extensive preprocessing [37]. The prediction of the ensemble at iteration m for an input vector x is
(5)$${\hat{y}}_{i}^{\left(m\right)}={\hat{y}}_{i}^{\left(m-1\right)}\;+\alpha \cdot f_{m}\left(x_{i}\right)$$
where ${\hat{y}}_{i}^{\left(m\right)}$ is the predicted value for observation i at iteration m. $f_{k}\left(x\right)$ is the prediction of the $m^{th}$ tree, and α is a learning rate. Distinctively, it utilizes an approach termed “ordered boosting” to mitigate overfitting, alongside a pooling technique to process categorical variables directly. Oblivious trees, a variant of decision trees with the same feature for splits at each level, form its foundational model structure, enhancing efficiency and reducing overfitting likelihood. Furthermore, the algorithm incorporates L2 regularization, further ensuring model robustness.

## 4. Experimental Results

### 4.1. Experimental Environments

In this study, vibration sensors were installed in water meter boxes and valve rooms to collect data for leak detection purposes. The water pipes used in the experiment were categorized into two types: metallic pipes and non-metallic pipes, and differences in vibration detection sensors were observed between them. Metallic pipes were found to transmit vibrations for a longer period, and clear signals could be obtained even at high frequencies. This characteristic makes leak detection in metallic pipes relatively accurate. On the other hand, non-metallic pipes are less effective in transmitting vibrations, and their frequency characteristics are different, making them more challenging to detect. Two types of vibration sensors were used in the experiment: LTE equipment and embedded equipment. These sensors were installed approximately 150 to 300 m apart in water meter boxes and valve rooms. Figure 8 shows the structure of the equipment used to collect the dataset for the leaky small-flow and water pressure monitoring system. Figure 9 displays the sensor models used for dataset collection and leak detection. The equipment and systems are utilized to detect vibrations in both metallic and non-metallic pipes to identify leaks accurately.

Water supply pipelines are essential infrastructure that transports water from a source to a destination. Various types of pipes are used in these pipelines, and they can be categorized into two types: metal pipes and non-metal pipes. Each type of pipe has different characteristics that affect its performance and accuracy of leak detection sensors.

Metal pipes are widely used in water supply pipelines. Enamel Coated Steel Pipe (ECSP) is highly resistant to corrosion, while Liquid Epoxy Coated Steel Pipe (LECSP) has improved corrosion prevention and durability. Cast Iron Pipe (CIP) is known for its heavy weight and high strength, while Ductile Iron Pipe (DIP) is popular for its improved toughness. Galvanized Steel Pipe (GSP) is a steel pipe with enhanced corrosion resistance, and Copper Pipe (CP) has excellent conductivity and heat resistance. Stainless Steel Pipe (SSP) has high corrosion resistance and durability, but its use is limited due to its high cost.

Non-metallic pipes are also used in water supply pipelines. Polyvinyl chloride (PVC) is lightweight and corrosion-resistant, while impact-resistant water pipe (IRWP) is impact-resistant. Polyethylene (PE) is corrosion-resistant and lightweight, while Hume Pipe (HP) is made of concrete and has a sturdy structure.

The different characteristics of water pipes affect the performance and accuracy of leak detection sensors. Therefore, it is necessary to evaluate and improve the performance of leak detection sensors by considering the type of water pipe. Table 3 shows the length (in meters) of water pipes in South Korea as of 2020, which can be used to evaluate the performance and applicability of leak detection sensors in each type of pipe.

### 4.2. Model Evaluation

In this study, we employed eight models. These include the nearest neighbor (KNN), decision tree, random forest, additive tree, and gradient boosting models from Scikit-learn (version 1.2.2) [38]. Additionally, we used LightGBM (version 3.3.5) [39], XGBoost (version 1.7.6) [40], and CatBoost (version 1.2) [41] from their individual libraries. The dataset was partitioned into 62,564 instances for the training phase and 7820 instances designated for evaluation. However, to optimize our experimental setup, we amalgamated both sets, resulting in a consolidated dataset of 70,384 instances. Subsequently, we reallocated the instances using an 8:2 split, aiming to augment the evaluation data volume. This restructuring yielded a training dataset of 56,307 instances, while the evaluation set encompassed 14,077 instances. For the modeling process, we employed multi-label classification techniques across all models. To ensure rigorous and comprehensive assessment, the training and evaluation phases were conducted employing a stratified K-Fold cross-validation approach, with a chosen K value of 10. In the present study, the target variable, denoted as “leaktype”, is categorically encoded as follows: ‘in’ is represented by the value 0, ‘noise’ is represented by the value 1, ‘normal’ is represented by the value 2, ‘other’ is represented by the value 3, and ‘out’ is represented by the value 4. In the present study, the experimental procedures are categorized into four distinct sections based on the features employed. A comprehensive overview of these divisions is provided in Table 4.

#### 4.2.1. Experimental Results for E1

For the E1 experiment, solely frequency-specific data derived from the vibration sensor were employed for leak detection as features. The outcomes of this experiment are delineated in Table 5. A perusal of Table 6 reveals that the K-Nearest Neighbors (KNN) model outperforms other models in terms of accuracy and computational efficiency. Figure 10 elucidates the classification efficacy of the KNN model. While the classes ‘in’, ‘normal’, and ‘out’—pertinent to the leakage status—demonstrate commendable performance, the ‘noise’ and ‘other’ classes exhibit suboptimal results.

#### 4.2.2. Experimental Results for E2

The E2 experiment incorporated the features from E1, augmented with the maximum number of leak detections, spanning 20 columns. The comparative performance of various models for this experiment is presented in Table 6. Intriguingly, the integration of aggregated maxima led to a decrement in the KNN model’s performance. Conversely, the random forest model emerged as the most efficacious. Figure 11 delineates the class-wise classification performance of the random forest model for the E2 experiment. Figure 12 underscores the significance of features for the random forest model, indicating a predilection of the model towards maximum value features introduced in the E2 experiment.

#### 4.2.3. Experimental Results for E3

The E3 experiment was characterized by the inclusion of leak rate and level. Over a span of 2 h during the early morning hours, 10 leak detection assessments were executed daily. Results surpassing a predefined threshold were represented as the leak probability (lrate), while the mean value of the responses was denoted as the leak level (llevel). Table 7 encapsulates the model performance metrics for the E3 experiment, with the random forest model exhibiting superior performance. Upon analysis of the experimental results, it is evident that there exists a modest enhancement in overall performance relative to the E2 experiment. Figure 13 shows the classification performance of the random forest model in the E3 experiment. Compared to the E2 experiment, there is a slight increase overall, except for class 3, “Other noise”.

Furthermore, Figure 14 accentuates the paramount importance of the newly incorporated leakage rate (lrate) in E3, suggesting that the random forest model predominantly classifies based on the leakage rate and maximum value features.

#### 4.2.4. Experimental Results for E4

For the E4 experiment, additional identification parameters were incorporated, namely the location of the leak detection sensor (site), the sensor’s unique identifier (sid), and the date of the leak detection event (ldate). Table 8 presents the performance metrics for the E4 experiment, highlighting the enhanced efficacy of both the tree and boost models, with the XGBoost model being preeminent. Figure 15 showcases the stellar classification performance of the XGBoost model for the E4 experiment.

Figure 16 delineates the feature importance for the XGBoost model in the E4 experiment, emphasizing the primacy of the leak rate (lrate), followed by the sensor number (sid). Notably, in contrast to the random forest model, the maximum value feature is of diminished importance in the XGBoost model.

## 5. Conclusions

In this study, our primary objectives revolved around extracting salient features from an expansive set of data, gathered via vibration sensors on various water pipe types under diverse leakage conditions. These data were then meticulously organized into a tabulated format, enabling us to identify and refine the most effective machine learning algorithm for water leak detection. Our evaluations revealed that the XGBoost model excelled in this domain, boasting a remarkable accuracy of 99.79% in detecting water leaks through vibration sensors. This achievement markedly surpasses conventional methods that depend on manual inspections or acoustic sensors, both of which are often susceptible to errors and ambient noise disruptions. Notably, our model adeptly identifies leaks in water pipelines composed of both metallic and non-metallic elements—a recurrent challenge in older water distribution systems.

The implications of our research are profound for the realm of water leak detection. By introducing a cutting-edge machine learning approach, we offer a potent solution, effectively addressing the limitations inherent in current detection techniques. The benefits of adopting this model are manifold, promising not only a significant reduction in water wastage but also a marked decrease in environmental impact, financial expenditure, and potential safety hazards stemming from unnoticed leaks. Moreover, the versatility of our proposed model allows its potential deployment in related sectors, including the gas, oil, and chemical industries. As we chart our future research path, our focus will shift towards optimizing the model for seamless execution on edge computing devices and delving deeper to ascertain its capability in pinpointing the root causes of water leaks.

## Figures and Tables

**Figure 1 sensors-23-08935-f001:**
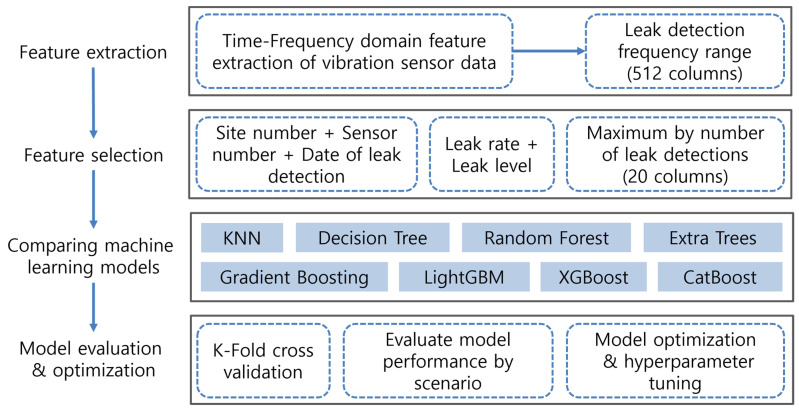
Flowchart of the proposed methodology.

**Figure 2 sensors-23-08935-f002:**
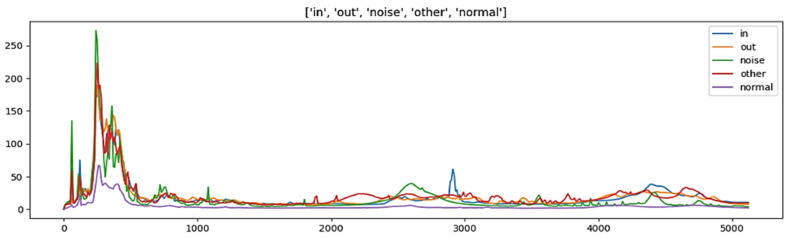
Leak detection data from 10 Hz to 5120 Hz.

**Figure 3 sensors-23-08935-f003:**
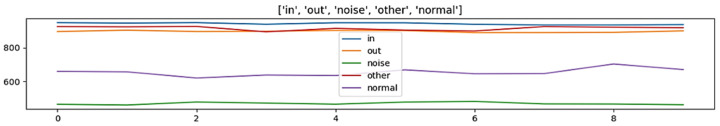
Maximum frequency and maximum leak size detected per leak detection (max 1~19).

**Figure 4 sensors-23-08935-f004:**
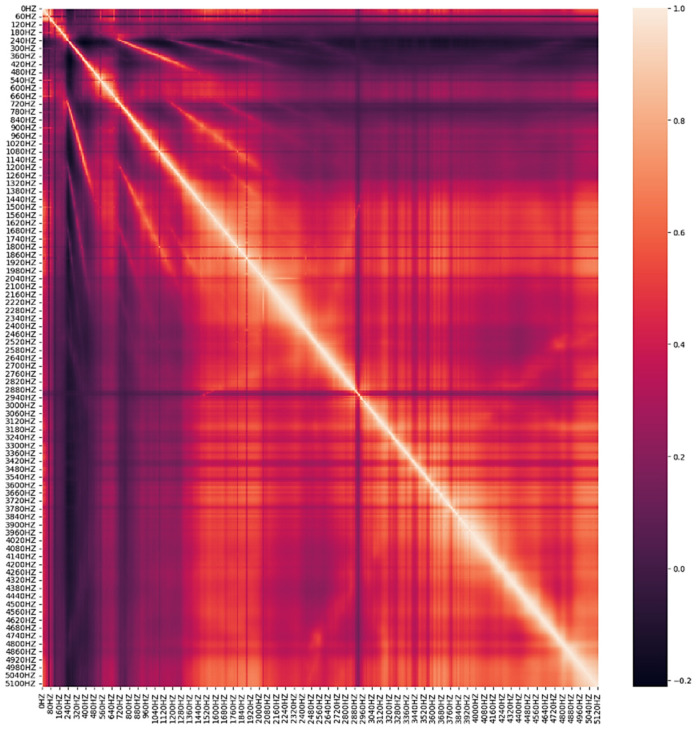
Correlation analysis results for leak detection data from 10 Hz to 5120 Hz.

**Figure 5 sensors-23-08935-f005:**
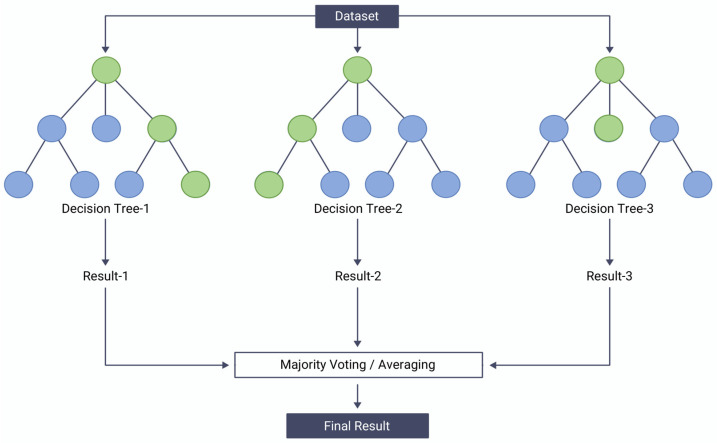
Example of random forest.

**Figure 6 sensors-23-08935-f006:**
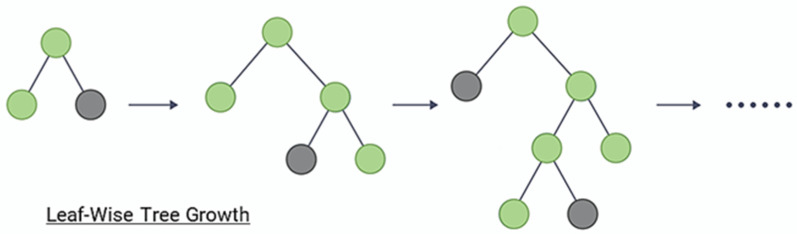
Example of leaf-wise method.

**Figure 7 sensors-23-08935-f007:**
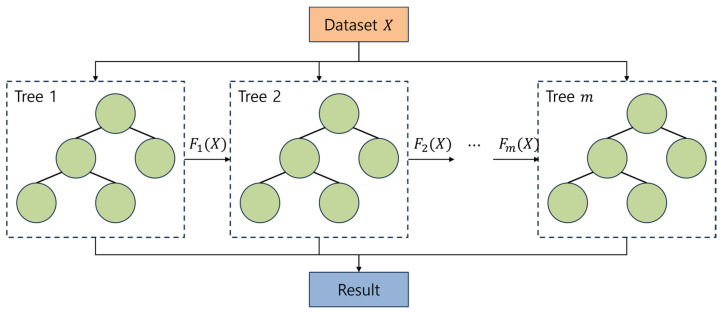
Example of XGBoost principle.

**Figure 8 sensors-23-08935-f008:**
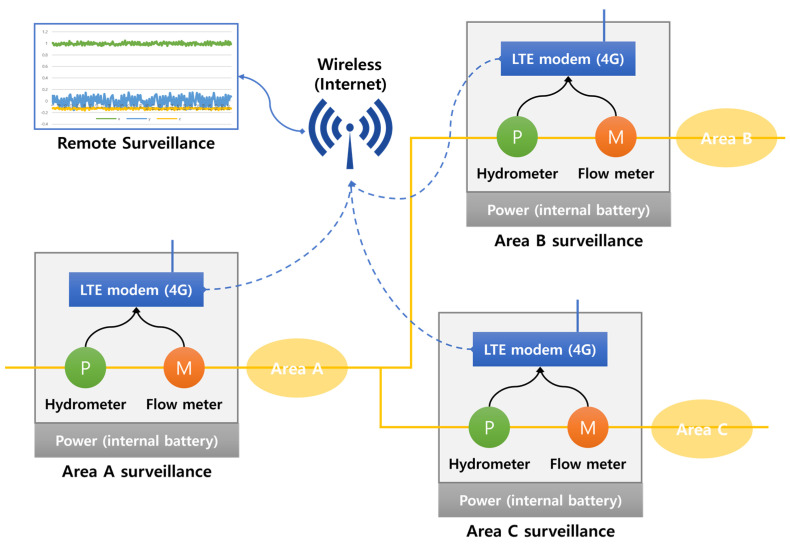
Flow and water pressure monitoring structure.

**Figure 9 sensors-23-08935-f009:**
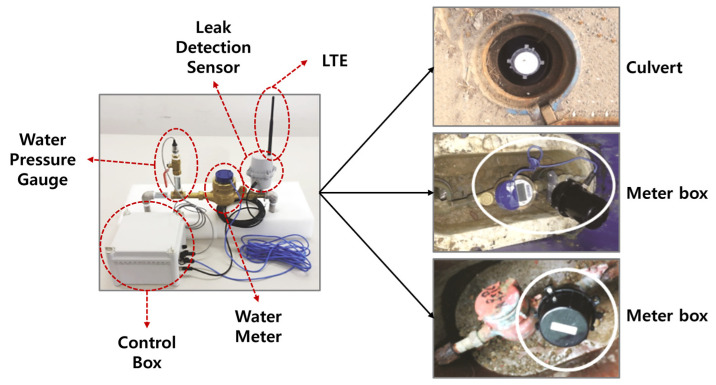
Flow and water pressure monitoring sensors.

**Figure 10 sensors-23-08935-f010:**
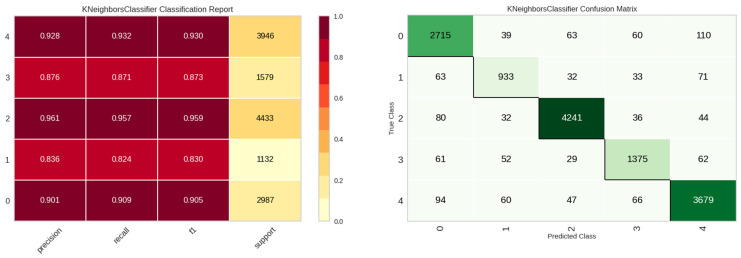
Classification performance results of the KNN model in E1 experiment.

**Figure 11 sensors-23-08935-f011:**
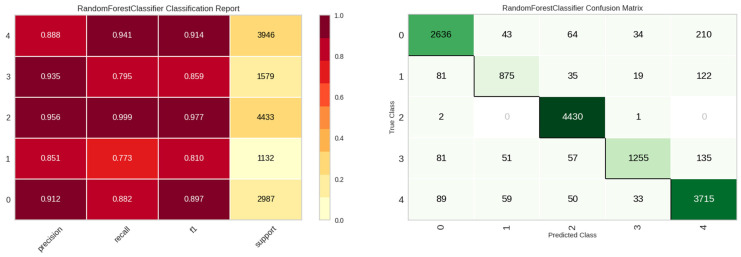
Classification performance results of the random forest model in E2 experiment.

**Figure 12 sensors-23-08935-f012:**
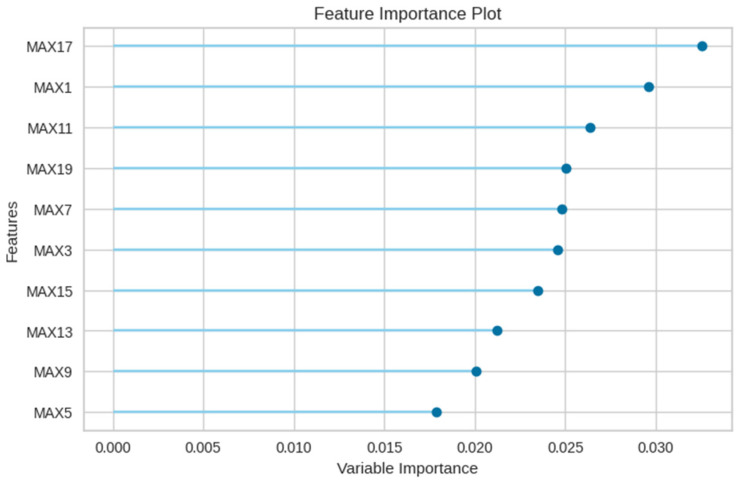
Feature importance of the random forest model in E2 experiment.

**Figure 13 sensors-23-08935-f013:**
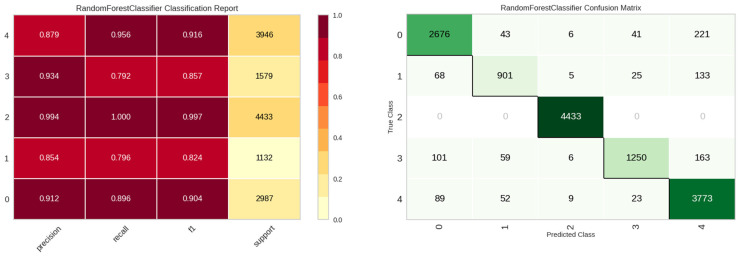
Classification performance results of the Random Forest model in E3 experiment.

**Figure 14 sensors-23-08935-f014:**
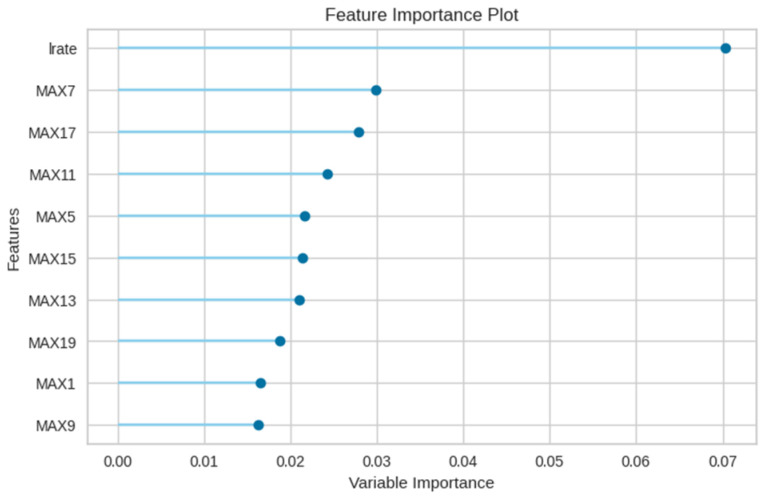
Feature importance of the random forest model in E3 experiment.

**Figure 15 sensors-23-08935-f015:**
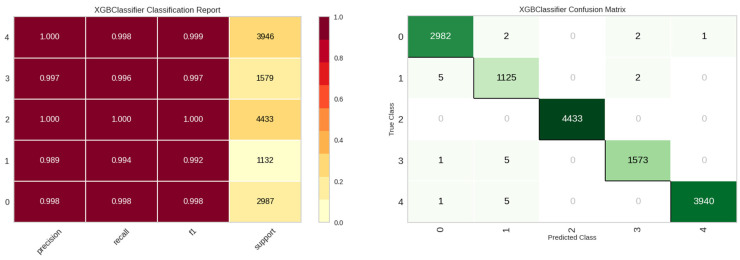
Classification performance results of the XGBoost model in E4 experiment.

**Figure 16 sensors-23-08935-f016:**
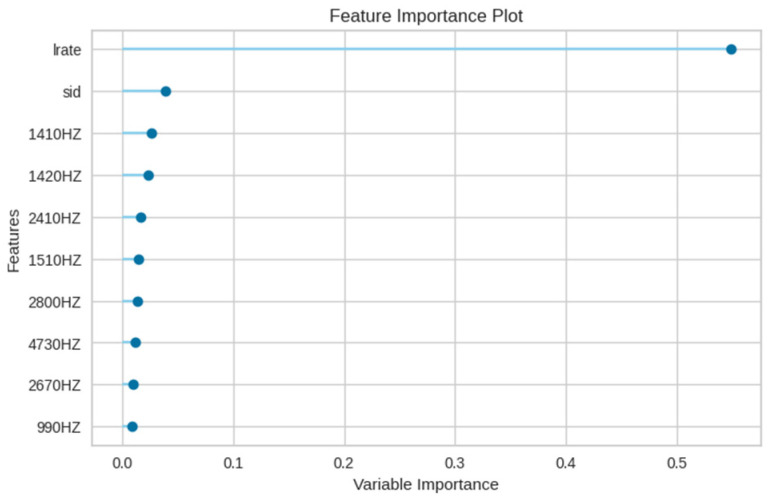
Feature importance of the XGBoost model in E4 experiment.

**Table 1 sensors-23-08935-t001:** Water pipeline leak detection dataset summary.

Water Pipe	Status Type	Leak Class	Train Data	Test Data
Metallic Pipes: Steel Pipe (SP), Stainless Steel Pipe (STS), Ductile Cast Iron Pipe (DCIP) Non-metallic Pipes: Polyethylene (PE), Polyvinyl chloride (PVC)	Leak detection	Outdoor leak (out)	17,539	2192
		Indoor leak (in)	13,273	1659
		Electric noise (noise)	5029	629
		Other noise (other)	7019	878
	Undetected	Normal noise (normal)	19,704	2462
Total	2	5	62,564	7820

**Table 2 sensors-23-08935-t002:** Format and description of water pipeline leak detection data.

Name	Column	Description	Format or Range	Type
Site no.	site	Site number where the leak detection sensor is installed (identifier 1)	S-00000000	string
Sensor no.	sid	Leak detection sensor number (identifier 2)	S-00000 00000 00000	string
Date of leak detection	ldate	The date the leak event was detected (identifier 3)	YYYYMMDD	string
Leak rate	lrate	Daily dawn hours for a total of 10 leak detection leak detection and set a schedule threshold over a certain threshold and display the result as a probability (Automatically create a leak detection sensor)	0~90	int
Leak level	llevel	Daily at dawn a total of 10 water leak detection leak detection, and if the threshold over a certain threshold and display the average value. (Automatically create a leak detection sensor)	0~5000	int
Leak Detection Classification	leaktype	Indicates whether a leak is detected.	outdoor leak (out), indoor leak (in), electric noise (noise), other noise (other), normal noise (normal)	string
Leak Detection Frequency Range (512 columns)	0 Hz	Leakage vibration magnitude detected at frequency 0 Hz	0	int
	10 Hz	Leakage vibration magnitude detected at frequency 10 Hz	0~5000	int
	…	…	…	…
	5120 Hz	Leakage vibration magnitude detected at frequency 5120 Hz	0~5000	int
Maximum by number of leak detections (20 columns)	MAX0	Maximum frequency Hz detected in one leak detection	0~5120	int
	MAX1	Maximum leak size detected at one leak detection	0~5000	int
	…	Maximum frequency Hz and maximum leak size detected in the number of leak detections	0~5000	int
	MAX18	Maximum frequency Hz detected in 10 leak detections	0~5120	int
	MAX19	Leak detection 10 times maximum leak size	0~5000	int

**Table 3 sensors-23-08935-t003:** Distances by water and sewer pipes in South Korea (2020).

Classification	Type	Distance (m)
Metallic Pipes	ECSP	975,242
	LECSP	9,983,024
	CIP	12,379,317
	DIP	56,089,910
	GSP	762,782
	CP	273,259
	SSP	25,565,017
Non-metallic Pipes	PVC	24,061,418
	IRWP	29,260,560
	PE	44,812,817
	HP	35,990
Other Pipes	OTH	16,023,203
Total		228,322,539

**Table 4 sensors-23-08935-t004:** Features used in the model per experiment.

No.	Features	Target
E1	Leak Detection Frequency Range (512 columns)	Leak Detection Classification (leaktype)
E2	Leak detection frequency range (512 columns) + Maximum by number of leak detections (20 columns)	
E3	Leak rate + Leak level + Leak detection frequency range (512 columns) + Maximum by number of leak detections (20 columns)	
E4	Site number + Sensor number + Date of leak detection + Leak rate + Leak level + Leak detection frequency range (512 columns) + Maximum by number of leak detections (20 columns)	

**Table 5 sensors-23-08935-t005:** Compare performance by model for the E1 experiment.

Model	Accuracy	AUC	Recall	Precision	F1	Kappa	MCC	Training Time
KNN	0.9133	0.9818	0.9133	0.9135	0.9133	0.8857	0.8857	2.1840
XGBoost	0.8917	0.9836	0.8917	0.8913	0.8903	0.8561	0.8567	41.6480
MLP	0.8788	0.9759	0.8788	0.8810	0.8789	0.8402	0.8407	19.7120
Random Forest	0.8780	0.9807	0.8780	0.8794	0.8755	0.8372	0.8387	2.8340
LightGBM	0.8765	0.9797	0.8765	0.8764	0.8748	0.8358	0.8366	8.3830
CatBoost	0.8708	0.9781	0.8708	0.8700	0.8684	0.8280	0.8290	60.2310
Extra Trees	0.8389	0.9724	0.8389	0.8443	0.8335	0.7834	0.7872	2.8490
Decision Tree	0.7705	0.8529	0.7705	0.7708	0.7706	0.6973	0.6973	2.2730
Gradient Boosting	0.6978	0.9089	0.6978	0.7016	0.6791	0.5899	0.5979	94.0720

**Table 6 sensors-23-08935-t006:** Compare performance by model for the E2 experiment.

Model	Accuracy	AUC	Recall	Precision	F1	Kappa	MCC	Training Time
Random Forest	0.9115	0.9892	0.9115	0.9113	0.9099	0.8823	0.8830	2.8770
XGBoost	0.9008	0.9860	0.9008	0.9017	0.9001	0.8685	0.8691	46.6050
LightGBM	0.8971	0.9850	0.8971	0.8981	0.8962	0.8635	0.8642	8.7430
CatBoost	0.8933	0.9846	0.8933	0.8928	0.8920	0.8585	0.8590	61.5220
Extra Trees	0.8655	0.9797	0.8655	0.8668	0.8611	0.8199	0.8221	2.7550
MLP	0.8472	0.9701	0.8472	0.8532	0.8478	0.7992	0.8004	24.8520
Decision Tree	0.8076	0.8789	0.8076	0.8079	0.8077	0.7463	0.7463	2.1420
KNN	0.7532	0.9201	0.7532	0.7532	0.7521	0.6738	0.6744	2.2280
Gradient Boosting	0.7360	0.9302	0.7360	0.7494	0.7231	0.6439	0.6526	100.9720

**Table 7 sensors-23-08935-t007:** Compare performance by model for the E3 experiment.

Model	Accuracy	AUC	Recall	Precision	F1	Kappa	MCC	Training Time
Random Forest	0.9221	0.9911	0.9221	0.9225	0.9212	0.8967	0.8972	2.5410
XGBoost	0.9030	0.9868	0.9030	0.9035	0.9023	0.8715	0.8720	38.0860
CatBoost	0.8972	0.9857	0.8972	0.8973	0.8963	0.8637	0.8642	61.6760
LightGBM	0.8962	0.9853	0.8962	0.8970	0.8954	0.8624	0.8630	8.6350
Extra Trees	0.8960	0.9859	0.8960	0.8969	0.8938	0.8616	0.8627	2.2420
MLP	0.8473	0.9723	0.8473	0.8533	0.8462	0.7981	0.8000	28.6360
Decision Tree	0.8321	0.8949	0.8321	0.8321	0.320	0.7785	0.7785	1.7550
KNN	0.7506	0.9185	0.7506	0.7507	0.7497	0.6705	0.6710	2.2210
Gradient Boosting	0.7369	0.9300	0.7369	0.7541	0.7239	0.6448	0.6547	99.8280

**Table 8 sensors-23-08935-t008:** Compare performance by model for the E4 experiment.

Model	Accuracy	AUC	Recall	Precision	F1	Kappa	MCC	Training Time
XGBoost	0.9979	1.0000	0.9979	0.9979	0.9979	0.9973	0.9973	37.9660
LightGBM	0.9971	1.0000	0.9971	0.9972	0.9971	0.9962	0.9962	8.7510
CatBoost	0.9950	0.9998	0.9950	0.9951	0.9950	0.9934	0.9934	60.6540
Gradient Boosting	0.9909	0.9999	0.9909	0.9912	0.9910	0.9881	0.9881	100.3840
Decision Tree	0.9878	0.9927	0.9878	0.9878	0.9878	0.9839	0.9839	1.3310
Random Forest	0.9828	0.9994	0.9828	0.9828	0.9827	0.9773	0.9774	2.4130
Extra Trees	0.9721	0.9987	0.9721	0.9721	0.9718	0.9631	0.9632	2.0980
KNN	0.8296	0.9541	0.8296	0.8296	0.8291	0.7751	0.7753	2.2800
MLP	0.4084	0.6336	0.4084	0.4084	0.3289	0.2402	0.2990	16.3160

## Data Availability

The Water pipe leak detection dataset is openly available at: https://www.aihub.or.kr/aihubdata/data/view.do?currMenu=115&topMenu=100&aihubDataSe=realm&dataSetSn=138 (accessed on 5 October 2023).

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
