# Peer review of "Machine Learning Model for Leak Detection Using Water Pipeline Vibration Sensor"

_sensors, 2023, doi:10.3390/s23218935_

Round 1

Reviewer 1 Report

Comments and Suggestions for Authors

Comments on the Quality of English Language

Moderate

Author Response

In this paper, the authors present ‘Machine Learning Model for Leak Detection Using Water Pipe- 2 line Vibration Sensor. Overall the presented discussion is very important in the field of underwater applications. Our comments are given below.

→ First of all, thank you for your valuable feedback.

  1. I recommend polishing the abstract section, including the novelty, methods, results, and findings.

→ Thank you for your valuable feedback. We have rewritten the summary to include the main contribution and novelty that we used tabular data separated by frequency in a form that can be trained on a machine learning model based on vibration sensors, and achieved a high performance of 99.79%.

  1. Revise the contributions up to a maximum of 3 points by following the results and discussions at the end of Section 1.

→ We appreciate your feedback, and we've adjusted it to three by removing one of the major contributions.

  1. It is recommended to strengthen the literature review section by writing a discussion about

recently published works in underwater sensor networks and its applications such as https://doi.org/10.3390/s19214762

https://doi.org/10.1016/j.adhoc.2019.101912

→ We've added the two papers you suggested, thank you for recommending them.

  1. Write a pseudo code of the proposed scheme and explain in the text in Section 4.

→ We have taken your advice and added a flowchart of the methodology we propose in the paper, and added more detail accordingly.

  1. Add key mathematical eqs. of Random Forest & Extra Trees algorithms.

→ Thanks for the advice. We have organized the formulas for each of the main algorithms, as well as the Random Forest & Extra Trees algorithms, and written them in the text.

  1. Figure 7 must be explained in detail. Some text is not seen clearly.

→ We redrew and inserted Figure 7 to make it clearer.

  1. Further improve the results and discussion by adding results in the context of what was known before and then discuss potential challenges, future directions/recommendations, and why these kinds of studies are important!

→ Thank you. We have rewritten the conclusion based on your advice.

Reviewer 2 Report

Comments and Suggestions for Authors

This study focuses on improving leak detection and response methods for aging water and wastewater pipes. Data from leak detection sensors at water meter boxes and pipelines were analyzed, and machine learning and deep learning models were developed and compared. The XGBoost model performed the best, achieving a 99.79% accuracy rate. This model can replace current methods, reducing detection and response time, conserving water, and minimizing economic losses.

The article addresses an important and intriguing issue, especially since the achieved results may have practical applications. It offers potential applications in various industries, promoting efficient resource management and sustainability. However, the reviewer has some reservations regarding the structure of the paper:

1)          The authors have opted for an unconventional arrangement of the individual chapters, which, in the reviewer's opinion, is not a suitable choice. The authors should consider reordering the text so that the introduction incorporates the literature review.

2)          I didn't find a reference to the literature source number 28 in the text.

3)          According to the reviewer, the article is written in good language, and it addresses interesting topics. However, there is a significant emphasis on the description of methods and their applications. This is a scientific article, and it should primarily present the results of the analyses conducted by the authors. Perhaps the authors should consider combining method descriptions with descriptions of the experiments conducted. Currently, the scientific content of this article indeed constitutes less than half of all the pages.

In summary, the authors should consider reworking the article and revisiting its structure.

Author Response

This study focuses on improving leak detection and response methods for aging water and wastewater pipes. Data from leak detection sensors at water meter boxes and pipelines were analyzed, and machine learning and deep learning models were developed and compared. The XGBoost model performed the best, achieving a 99.79% accuracy rate. This model can replace current methods, reducing detection and response time, conserving water, and minimizing economic losses.

The article addresses an important and intriguing issue, especially since the achieved results may have practical applications. It offers potential applications in various industries, promoting efficient resource management and sustainability. However, the reviewer has some reservations regarding the structure of the paper:

→ First of all, thank you for seeing the many benefits and potentials of this paper.

The authors have opted for an unconventional arrangement of the individual chapters, which, in the reviewer's opinion, is not a suitable choice. The authors should consider reordering the text so that the introduction incorporates the literature review.

→ We've taken your advice and rearranged the order of the text, and made some major changes to the overall layout.

I didn't find a reference to the literature source number 28 in the text.

→ Thank you for your thoughtful review. We've removed the reference.

According to the reviewer, the article is written in good language, and it addresses interesting topics. However, there is a significant emphasis on the description of methods and their applications. This is a scientific article, and it should primarily present the results of the analyses conducted by the authors. Perhaps the authors should consider combining method descriptions with descriptions of the experiments conducted. Currently, the scientific content of this article indeed constitutes less than half of all the pages.

→ Thank you for your valuable comment. We've made significant changes that cut to the chase. We've expanded the description of our methodology and step-by-step process, detailed the experimental setup and results, and added a number of formulas and illustrations for the reader.

In summary, the authors should consider reworking the article and revisiting its structure.

→ We have rewritten the paper and changed its structure in response to your considerations. We believe that your advice has made the paper more informative for our readers.

Reviewer 3 Report

Comments and Suggestions for Authors

The Authors performed a comparison of the efficiency of several data classification methods/approaches in the task of leak detection in water pipelines. However, in this version of the submission, it is not possible to evaluate the to assess the usability of this comparison, as:

  -- no information about the eperiment setup is provided (if they detect leakages from a single pipe or from a water network of a certain configuration, what are the physical dimensions of the pipe/network, the location of sensors, etc., etc.);

  -- no data characteristics are provided, e.g. if the dataset used in experiments is available or was acquired by the Authors (and how), what the terms: "Outdoor leak", "Indoor leak", etc. stand for (i.e. what the typical data characteristics for each class are, what "sizes" of leakages where considered, etc.).

Concluding the above - quality of presentation is definitely the weakest side of the submission. Additionally - the technique of using references must be also improved, e.g. the source of data presented in Table 1 must be given, the book/paper presenting the kNN method as well, as in some other places where the Autors describe the work of other researchers, commonly available. Also the content of most of the figures must be commented in the text. For example, the content of Figure 3 does not help to understand the concept of usage of the decision tree method.

Author Response

The Authors performed a comparison of the efficiency of several data classification methods/approaches in the task of leak detection in water pipelines. However, in this version of the submission, it is not possible to evaluate the to assess the usability of this comparison, as:

  -- no information about the eperiment setup is provided (if they detect leakages from a single pipe or from a water network of a certain configuration, what are the physical dimensions of the pipe/network, the location of sensors, etc., etc.);

→ Thank you for your feedback. The water supply pipes are a mix of metal pipes (SP, STS, DCIP) and non-metal pipes (PE, PVC), the leak detection range is more than 150 meters for drainage pipes and more than 70 meters for water supply pipes, and the sensors are installed in the water meter box and valve room respectively. We have added a section on the experimental environment to describe it.

  -- no data characteristics are provided, e.g. if the dataset used in experiments is available or was acquired by the Authors (and how), what the terms: "Outdoor leak", "Indoor leak", etc. stand for (i.e. what the typical data characteristics for each class are, what "sizes" of leakages where considered, etc.).

→ Your advice was very helpful. The dataset used in our experiments is a public version, and we have written the URL in the paper. The dataset was created with the support of the Korean government, with the participation of leak detection companies and experts. It was created for real-world smart leak detection. The following figure shows some sections of the real-world environment included in the dataset.

We added a detailed description of each class in the text. We also detailed the characteristics of the dataset we used.

Concluding the above - quality of presentation is definitely the weakest side of the submission. Additionally - the technique of using references must be also improved, e.g. the source of data presented in Table 1 must be given, the book/paper presenting the kNN method as well, as in some other places where the Autors describe the work of other researchers, commonly available. Also the content of most of the figures must be commented in the text. For example, the content of Figure 3 does not help to understand the concept of usage of the decision tree method.

→ Following your advice, we have improved the quality of the presentation: we have added more figures and formulas for the reader, changed the text, added references to the kNN method, and added explanations to the figures in the text.

Round 2

Reviewer 1 Report

Comments and Suggestions for Authors

Accepted 

Author Response

Thank you for your valuable acceptance.

Reviewer 3 Report

Comments and Suggestions for Authors

The Authors improved teh quality of presentation, following the comments and remarks of the reviewer. It mostly helped, however there are still some unclear and/or ambiguous parts of the text.

For example - when data preprocessing stage is described, there is a sentence (lines 184-186): "Subsequent to this extraction, metadata such as site number, sensor number, and the date of leak detection were incorporated to fortify the identification framework.' Does it mean that the metadata was also passed to the classifier inputs? How?

Also figure 5 does not illustrate anything specific for the classification task and may be excluded.

I also think that placing paragraphs between lines 399 to 418 at the beginning of page #13 is not a good idea.

Author Response

The Authors improved teh quality of presentation, following the comments and remarks of the reviewer. It mostly helped, however there are still some unclear and/or ambiguous parts of the text.

→ Thank you for your advice, your feedback has made our paper better.

For example - when data preprocessing stage is described, there is a sentence (lines 184-186): "Subsequent to this extraction, metadata such as site number, sensor number, and the date of leak detection were incorporated to fortify the identification framework.' Does it mean that the metadata was also passed to the classifier inputs? How?

→ As you astutely observed, I realize that I may not have been explicit regarding the features incorporated as input to the model. We have amended the text to provide greater clarity on the additional features employed as inputs to the model.

Also figure 5 does not illustrate anything specific for the classification task and may be excluded.

→ As per your comment, we have removed Figure 5 from the text.

I also think that placing paragraphs between lines 399 to 418 at the beginning of page #13 is not a good idea.

→ We rearranged the placement of figures and paragraphs throughout to improve the quality of the paper.
